# Estimating 24-Hour Sodium Excretion from Spot Urine Samples in Chinese Adults: Can Spot Urine Substitute 24-Hour Urine Samples?

**DOI:** 10.3390/nu12030798

**Published:** 2020-03-18

**Authors:** Jianwei Xu, Jiyu Zhang, Min Liu, Yamin Bai, Xiaolei Guo, Jing Dong, Aiqiang Xu, Jing Wu

**Affiliations:** 1National Center for Chronic and Noncommunicable Disease Control and Prevention, Chinese Center for Disease Control and Prevention, Beijing 100050, China; xujianwei@ncncd.chinacdc.cn (J.X.); liumin@ncncd.chinacdc.cn (M.L.); baiyamin@ncncd.chinacdc.cn (Y.B.); 2Shandong Center for Disease Control and Prevention, Jinan 250014, China; zhangjiyu008@163.com (J.Z.); guoxiaolei@126.com (X.G.); freedom.dj1@163.com (J.D.); aqxuepi@163.com (A.X.)

**Keywords:** 24-hour urine, spot urine, urinary sodium excretion

## Abstract

Several estimating equations for predicting 24-h urinary sodium (24-hUNa) excretion using spot urine (SU) samples have been developed, but have not been readily available to Chinese populations. We aimed to compare and validate the six existing methods at population level and individual level. We extracted 1671 adults eligible for both 24-h urine and SU sample collection. Mean biases (95% CI) of predicting 24-hUNa excretion using six formulas were 58.6 (54.7, 62.5) mmol for Kawasaki, −2.7 (−6.2, 0.9) mmol for Tanaka, −24.5 (−28.0, −21.0) mmol for the International Cooperative Study on Salt, Other Factors, and Blood Pressure (INTERSALT) with potassium, –26.8 (−30.1, −23.3) mmol for INTERSALT without potassium, 5.9 (2.3, 9.6) mmol for Toft, and −24.2 (−27.7, −20.6) mmol for Whitton. The proportions of relative difference >40% with the six methods were nearly a third, and the proportions of absolute difference >51.3 mmol/24-h (3 g/day salt) were more than 40%. The misclassification rate were all >55% for the six methods at the individual level. Although the Tanaka method could offer a plausible estimation for surveillance of the population sodium excretion in Shandong province, caution remains when using the Tanaka formula for other provincial populations in China. However, these predictive methods were inadequate to evaluate individual sodium excretion.

## 1. Introduction

The association of high salt intake with blood pressure (BP) has been widely accepted. Excessive salt intake can increase the risk of stroke and cardiovascular disease [1,2]. A diet high in sodium was one of the top three risk factors for both deaths and disability-adjusted life-years in China [3]. Mean population salt intake in China was 12–14 g/day [4], which is more than twice the World Health Organization (WHO) recommended salt intake (<5.0 g/day). Hence, salt reduction is a critical public health priority to prevent hypertension in China. WHO has recommended a 30% relative reduction in mean population salt intake by 2025. The Chinese central government targets 20% relative reduction in mean population salt intake by 2030, which is one of the key targets of Healthy China 2030. It is essential that the mean population sodium intake can be continuously monitored and tracked to meet the salt reduction target. Therefore, accurate measurements of salt intake is essential to evaluate the effectiveness of salt reduction.

There are several different methods commonly used to estimate salt intake, including food frequency questionnaires, 24-h dietary recall, and 24-h urinary sodium (24-hUNa) excretion. [5]. Until now, 24-hUNa excretion has been the most accurate method for estimating salt intake [6]. However, urine collection over a 24-h period is troublesome for participants to adhere to and researchers to conduct in epidemiological surveys. Low response rates and incomplete 24-h urine collection often occur in large-scale population surveys [7,8]. In order to solve these practical problems, some researchers have developed several formulas that may estimate 24-hUNa excretion using spot urine (SU) samples. SU sample was easier and better accepted by participants. These methods include the Kawasaki formula [9], Tanaka formula [10], INTERSALT with potassium formula (INTERSALT1) [11], INTERSALT without potassium formula (INTERSALT2) [11], Toft formula [12], and the Whitton formula [13]. These equations based on SU samples have many advantages and give hope for estimating population salt intake. This is particularly important for China, which is implementing country-level salt reduction strategies. Unfortunately, recent validation studies in the Chinese population remains controversial because the results of different studies are inconsistent. To date, whether these formulas based on SU samples may be a reasonable estimated method for Chinese population remains a key unresolved issue. Consequently, suitability and accuracy of estimation formulas need more research in the larger Chinese population.

This study aims to explore whether SU samples can perform well to estimate sodium intake both at population level and at individual level in China, with a view to evaluate whether SU samples can substitute 24-h urine samples.

## 2. Materials and Methods

### 2.1. Study Participants

The article utilized participants from the final evaluation survey of the Shandong–Ministry of Health Action on Salt and Hypertension project (SMASH). The methods and preliminary results for the SMASH project have been previously published [14,15]. The final survey was conducted in Shandong province in 2016. In included 2184 subjects aged 18–69 years were invited and asked to collect SU samples and 24-h urine samples. Finally, 2043 participants completed 24-h urine samples with a participation rate at 93.5%. We excluded 368 ineligible 24-h urine samples and 4 participants missing SU samples. The eligibility of the 24-h urine samples was judged with reference to US National Institutes of Health GenSalt’s salt sensitivity-related standards. If the collection time was less than 22-h, more than two urine samples were not collected, a participant recall regarding timing or missed collections was uncertain, the urine volume was <500 mL, or urinary creatinine was not within ±2 standard deviation of the gender-specific mean, then 24-h urine samples were deemed ineligible [15]. The final analysis sample included 1671 participants.

All participants provided written informed consent. The study complied with the Declaration of Helsinki guidelines, and any procedures involving human subjects were approved by the ethics committee of the National Center for Chronic and Noncommunicable Disease Control and Prevention, Chinese Center for Disease Control and Prevention.

### 2.2. Urine Collection and Laboratory Testing

Each respondent was provided with a plastic bottle with a cap for 24-h urine sample collection, which contained around 1.0 g of boric acid. Each respondent was given a urine collection cup and a storing SU sample tube. Participants had the procedure explained verbally and in writing for handling the SU and 24-h urine samples. One clean midstream SU sample was collected in the morning at home. Next, participants came to the investigation site with the SU sample. The 24-h urine collection began after participants emptied their bladders at the investigation site. Subsequently, participants collected all urine samples over the next 24-h. Each participant had to return to the investigation site at the same time the second day. The last urine had to be collected into the bottle. The starting and ending time of the 24-h urine collection were recorded by a supervising health professional. A questionnaire assessing the completeness of urine collection was verbally asked to each respondent after participants completed urine collection.

The 24-h urine samples from each participant were carefully stirred and the total volume was measured. A 20 mL aliquots of urine was frozen at −20 °C. All urine samples were transported frozen to the ADICON Clinical Laboratory Inc., Jinan, China, and were analyzed. The 24-hUNa and urinary potassium tests were performed with the ion-selective electrode method using the Olympus AU 680 autoanalyzer (Beckman Coulter k.k., Tokyo, Japan). Urine creatinine was tested with the enzyme method using the Olympus AU 640 analyzer (Beckman Coulter k.k., Tokyo, Japan).

### 2.3. Other Measurements

Other measurements included height, weight and BP. Participants were advised to wear light indoor clothing without shoes before the height and weight examination. The body-mass index (BMI) was calculated according to the formula: weight (kg)/height (m)^2^. BP was measured using a certified automated device (HEM-7071, Omron Corporation, Dalian, China). Three readings were recorded at one-minute intervals. The average of the three measurements was used.

### 2.4. Estimation of 24-hUNa Excretion

24-hUNa excretion was predicted from SU samples using six existing estimation methods: Kawasaki, Tanaka, INTERSALT1, INTERSALT2, Toft, and Whitton. These formulas estimate 24-hUNa excretion based on age, height, weight, and urinary sodium and creatinine (and potassium for INTERSALT1) in SU samples. The formulas are shown in Appendix A.

### 2.5. Statistical Analysis

Participant characteristics were presented by gender, and differences between male and female subjects were tested by independent samples *T*-test. At population level, we firstly calculated estimated values of 24-hUNa excretion for each of the six formulas. Next, the mean bias were calculated by predicted values minus measured values of 24-hUNa excretion. Differences between predicted and measured values of 24-hUNa excretion were tested with paired samples *T*-test. Pearson’s *r*, intraclass correlation coefficients (ICC), and scatter plots were used to assess the relation between estimated 24-hUNa from six estimation formulas and measured 24-hUNa excretion. The Bland–Altman plots were used to evaluate the agreement between measured and estimated 24-hUNa excretion.

At individual level, the relative differences were calculated by the following formula: (predicted value-measured value/measured value). Absolute differences were analyzed according to the formula: (absolute of (predicted value-measured value)). Salt intake was calculated from estimated and measured 24-hUNa and classified into 4 categories (<9, 9–11.99, 12–14.99, and ≥15 g/day), which were easy to compare with other studies. Next, we calculated the misclassification rates. All data were analyzed using SAS 9.3 (SAS Institute Inc., Cary, NC, USA). Tests for differences were two-sided and *p* < 0.05 was set as the significance level.

## 3. Results

### 3.1. Participant Characteristics

A total of 1671 participants were included in this final analysis. The mean age of the subjects was 43.8±12.4 years and subjects were 50% male. Mean 24-h urine volume was 1545.2 mL. The measured mean 24-hUNa excretion was 176.4 ± 79.1 mmol/24-h, corresponding to a calculated salt intake of 10.3 ± 4.6 g/day. The characteristics of the participants are presented by gender in Table 1.

### 3.2. Measured Versus Estimated 24-hUNa at Population Level

The mean bias between predicted and measured 24-hUNa excretion were showed in Table 2. The mean bias for Tanaka formula was −2.7 mmol (95% CI: −6.2, 0.9 mmol), and was the smallest difference among the six predicted formulas. The largest mean bias were 58.6 mmol for Kawasaki. Using the Tanaka formula, mean estimated 24-hUNa excretion was not significantly different compared to measured values (*t* = −1.47, *p* = 0.1421). However, using the other five methods, mean predicted 24-hUNa excretion was significantly different compared to measured values (all *p* < 0.01).

Correlation analysis indicated that predicted 24-hUNa excretion using six methods were moderate positive correlations with measured 24-hUNa excretion. Pearson’s *r* between measured and estimated 24-hUNa excretion were moderate correlation (all *p* < 0.01). The highest Pearson’s *r* was 0.410 for INTERSALT1 and the lowest Pearson’s *r* was 0.374 for Toft. For scatter plots, see Appendix A. ICC were also moderate (all *p* < 0.01). The highest ICC was 0.58 for Kawasaki, and the lowest ICC was 0.47 for Tanaka and INTERSALT2.

Bland–Altman plots showed that the Tanaka formula performed relatively accurately among the six formulas (Figure 1), which also showed that an overestimation of 24-hUNa excretion occurred at low levels of sodium excretion, whereas an underestimation occurred at high levels of sodium excretion. Compared with other formula, this trend was relatively inconspicuous in the Kawasaki formula.

### 3.3. Measured Versus Estimated 24-hUNa at Individual Level

The measured 24-hUNa excretion was used as the reference. The highest proportions of relative differences within ±10% were 20.6% for Tanaka, and the proportions were less than 20% for the other five formulas (Figure 2). The proportions of relative differences beyond ±40% were all >30%, and the highest proportion was 51.0% for Kawasaki. The proportions of absolute difference within 17.1 mmol/24-h (1 g/day salt) for the six formulas were 13.5%, 20.7%, 20.7%, 20.2%, 19.2%, and 20.0%, respectively. Moreover, the proportions of absolute difference beyond 51.3 mmol/24-h (3 g/day salt) for six methods were 64.2%, 44.6%, 43.2%, 43.0%, 47.8%, and 45.0%, respectively (Figure 3).

After dividing the estimated and measured 24-hUNa excretion into four groups (<9 g/day, 9–11.99 g/day, 12-14.99 g/day, ≥15 g/day), the rates of individual misclassification of salt intake groups were all >50%, with 71.8% for Kawasaki, 60.4% for Tanaka, 56.3% for INTERSALT1, 57.0% for INTERSALT2, 60.9% for Toft, and 56.7% for Whitton (Table 3).

Additional analysis was performed. Salt intake was classified into another four categories: <7 g/day, 7–9.99 g/day, 10–12.99 g/day, and ≥13 g/day. The rates of individual misclassification of salt intake groups were all >60% (see Appendix A).

## 4. Discussion

In the current study, we aimed to validate the six predictive equations in estimating 24-hUNa excretion using first-morning SU samples at both population and individual levels. We found that the Tanaka formula showed the best performance in estimating population mean 24-hUNa excretion. INTERSALT1, INTERSALT2, and Whitton methods significantly underestimated population mean 24-hUNa excretion. By contrast, Kawasaki and Toft equations significantly overestimated. However, all these six methods performed poorly at individual levels. The misclassification rates at individual level were all high in the six methods.

Overall, of the six equations examined, Tanaka formula performed best and had most agreement with measured 24-hUNa excretion (mean bias: −2.7 mmol/24-h, 95%; Cl: −6.2, 0.9 mmol/24-h) at population level. In contrast, using the other five methods, the estimated population mean 24-hUNa excretion had significant biases (all *p* < 0.01). Two previous studies reported that Tanaka performed well to predict mean population 24-hUNa excretion in both Chinese adults [16] and young adolescents [17], which was in conformance with our study. Nevertheless, other previous studies in Chinese adults found that the Kawasaki formula showed the lowest bias compared to Tanaka and INTERSALT formulas in both general [18,19,20,21] and hypertensive populations [22]. However, the Kawasaki formula performed worst among the six equations in our study. It is possible that the reason could be explained because all six methods were not developed from the Chinese population. For instance, the Kawasaki formula was developed from the Japanese population and the Toft formula was developed from the Danish population. In addition, we used the first-morning SU to assess 24-hUNa excretion, rather than the second morning urine for the Kawasaki formula.

Although the Tanaka formula performed best in predicting 24-hUNa excretion at population level among these six methods, the correlation coefficient of this method was also moderate (0.378). All the correlation of the six equations were moderate (r = 0.374–0.410, all *p* < 0.01; ICC = 0.47–0.58, all *p* < 0.01) in our study. Correlation coefficients in the present results were better compared with previous studies in Chinese adults. Bland–Altman plots of the six methods showed that these equations may overestimate a lower sodium intake and underestimate a higher level of sodium intake, which was consistent with a study conducted by Zhang et al. [16]. A meta-analysis also found similar trends [23]. Recently, a cohort study suggested that estimating salt intake using SU samples may observe change in salt intake at population level over time in Australian adults [24]. Therefore, further study into larger and different populations is warranted.

Our study found that all these six methods performed poorly at individual levels in the current population. The proportion of relative differences beyond ±40% was quite large for all six methods. Similarly, the proportions of absolute difference beyond 3 g/day salt was quite large. Although the Tanaka formula showed the best performance in predicting 24-hUNa excretion at population level, the misclassification rates at individual level were more than 60%. These results were consistent with previous studies conducted in Chinese [17,20] and in US adults with chronic kidney disease [25]. It indicated that these six formulas might be inappropriate to estimate 24-hUNa excretion at individual level.

The WHO has set voluntary targets for mean population intake to reduce by a 30% relative reduction by 2025. Because of high salt intake in China, it is very difficult for China to achieve this goal. Therefore, China has set a target for mean population salt intake by a 20% relative reduction for 2030, which is one of the key targets of Healthy China 2030. In 2019, Healthy China Action (2019–2030) was issued, which is a new guideline to promote people’s health through taking targeted measures and implementing 15 specific programs. Salt reduction action is an important initiative. We need to monitor change in Chinese population salt intake over time to evaluate the effectiveness of salt reduction programs. However, obtaining complete 24-h urine samples to estimate salt intake is challenging. Our study result showed SU samples may be a plausible method to substitute 24-h urine collection for surveillance population salt intake, although it is unacceptable for estimating individual salt intake. However, these formulas have inconsistent results in the Chinese population. The salt intake and dietary habits of people living in different regions of China vary widely [26], and this affects the accuracy of these formulas. Therefore, it might be difficult to establish a unified formula to estimate population mean salt intake in China.

Our study has two major strengths. To our knowledge, this is a reasonably large sample size and high participation rate of Chinese adults to validate the six methods in estimating 24-hUNa excretion using SU samples. In addition, we conducted strict quality controls in the 24-h urine collection step, and participants with incomplete or suspected incomplete urine collections were excluded during statistical analysis. However, there are several limitations to this study. Firstly, a single 24-hUNa excretion was not sufficient to evaluate the day-to-day variation in sodium excretion. Multiple consecutive 24-h collections were required in order to truly evaluate usual salt consumption [27]. In addition, 24-h urine samples with one occasional missing collection were eligible, and the duration of urine collection between 22 to 24 h were acceptable in the study, without any other adjustment. Secondly, we only collected the first morning spot urine to estimate 24-hUNa excretion, but the first morning urine has lower sodium and potassium concentrations compared with spot urine samples from other times [28]. Moreover, the Kawasaki equation was developed with second morning SU samples. Thirdly, our study just included one province in China to predict 24-hUNa excretion using SU samples, thus limiting the generalizability of our result. Therefore, more validating research in other provinces in China were needed.

## 5. Conclusions

In summary, we found the Tanaka formula could offer a plausible estimation for surveillance of the population’s sodium excretion in Shandong province. However, caution remains when using the Tanaka formula for other provincial populations in China, and further research into different populations is needed. These predictive methods were inadequate to evaluate 24-hUNa excretion at individual levels, and the 24-h urine collection remains the gold standard for individual salt intake.

## Figures and Tables

**Figure 1 nutrients-12-00798-f001:**
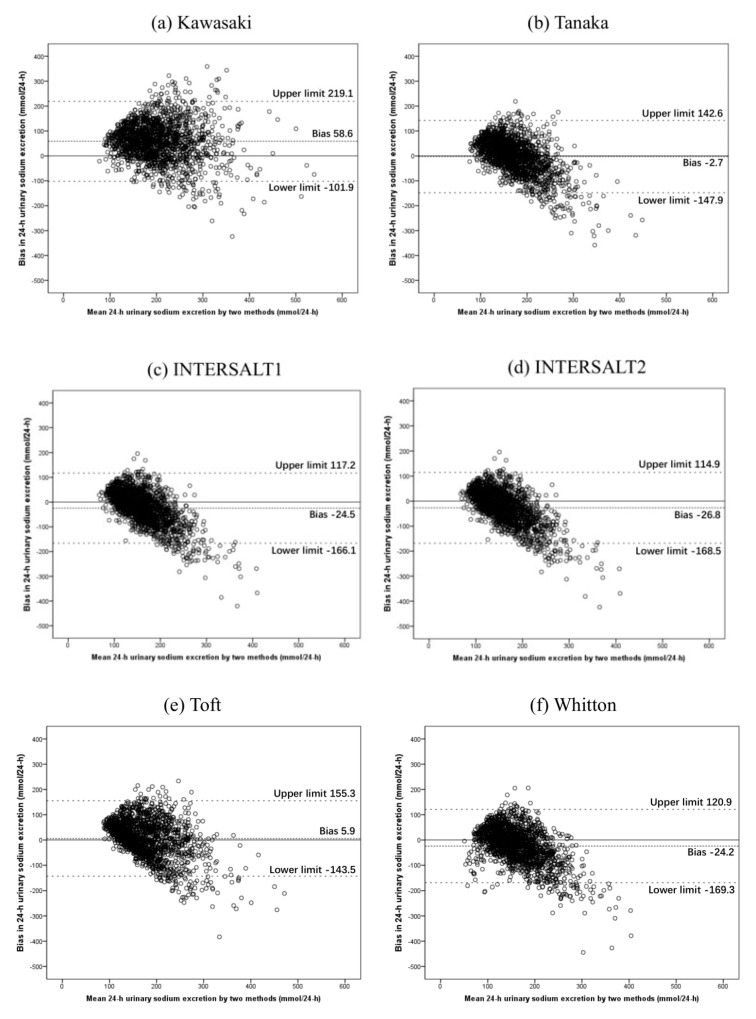
Bland–Altman plots comparing measured vs. estimated 24-hUNa excretion. (**a**) Kawasaki; (**b**) Tanaka; (**c**) INTERSALT1; (**d**) INTERSALT2; (**e**) Toft; (**f**) Whitton. The mid-dashed line was the mean bias. The dash-point line were 95% limits of agreement calculated by mean bias±1.96 standard deviation.

**Figure 2 nutrients-12-00798-f002:**
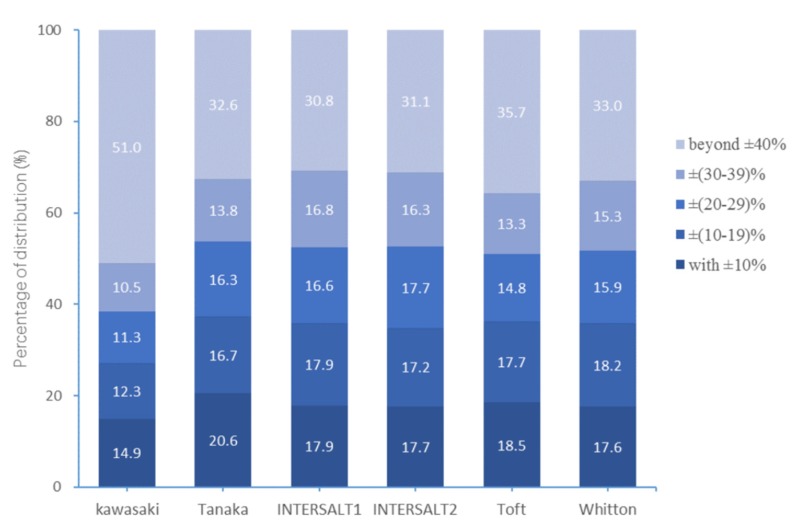
Relative difference distributions of measured and predicted 24-hUNa excretion.

**Figure 3 nutrients-12-00798-f003:**
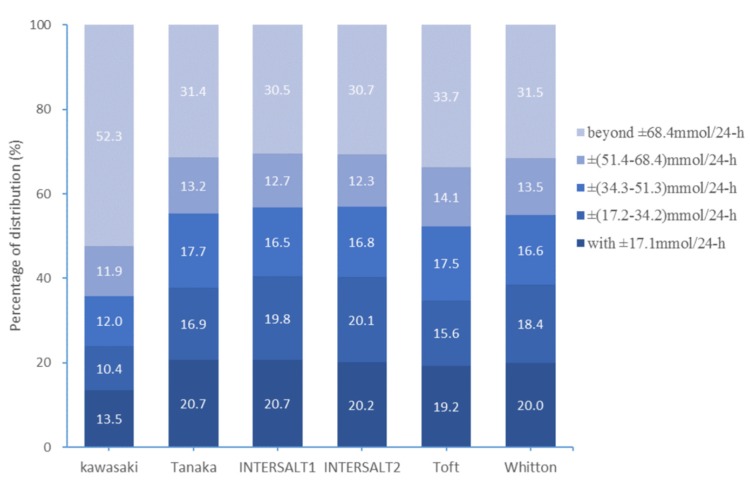
Absolute difference distributions of measured and predicted 24-hUNa excretion.

**Table 1 nutrients-12-00798-t001:** Characteristics of the 1671 participants according to gender, mean, and standard deviation.

	All (*n* = 1671)	Male (*n* = 836)	Female (*n* = 835)	*p*-Value ^1^
Age (years)	43.8 ± 12.4	42.6 ± 12.6	44.9 ± 12.1	0.0001
Weight (kg)	68.1 ± 13.6	73.3 ± 13.8	62.9 ± 11.2	<0.0001
Height (cm)	163.3 ± 8.6	169.2 ± 6.6	157.4 ± 5.8	<0.0001
BMI (kg/m^2^)	25.5 ± 4.2	25.6 ± 4.2	25.4 ± 4.2	0.4244
Systolic BP (mm Hg)	121.8 ± 17.9	124.8 ± 15.8	118.8 ± 19.3	<0.0001
Diastolic BP (mm Hg)	77.7 ± 11.7	79.2 ± 11.2	76.2 ± 12.1	<0.0001
Spot urine				
Sodium concentration (mmol/L)	135.5 ± 60.8	144.8 ± 61.2	126.3 ± 58.9	<0.0001
Potassium concentration (mmol/L)	32.0 ± 22.2	32.5 ± 21.9	31.6 ± 22.6	0.4385
Creatinine concentration (mmol/L)	9.2 ± 5.8	10.8 ± 6.1	7.7 ± 5.0	<0.0001
24-h urine				
24-h urine volume (mL)	1545.2 ± 599.9	1559.3 ± 614.3	1531.1 ± 585.2	0.3377
sodium excretion (mmol/24-h)	176.4 ± 79.1	190.3 ± 85.9	162.5 ± 69.0	<0.0001
potassium excretion (mmol/24-h)	47.3 ± 21.8	47.0 ± 23.2	47.5 ± 20.4	0.6196
Creatinine excretion (mmol/24-h)	10.5 ± 3.6	12.1 ± 3.6	8.9 ± 2.7	<0.0001

Body-mass index (BMI), blood pressure (BP), ^1^ Differences between male and female were tested by independent samples T-test.

**Table 2 nutrients-12-00798-t002:** Comparison between measured 24-hUNa excretion, predicted using six formulas (*n* = 1671).

	24-hUNa (mmol/24-h)	Mean Bias (mmol/24-h, 95% CI)	*p*-Value of Mean Bias	Person’s r ^1^	ICC (95%CI) ^2^
Estimated	176.4 ± 79.1				
Kawasaki	235.0 ± 70.7	58.6 (54.7, 62.5)	<0.0001	0.407	0.58 (0.53, 0.62)
Tanaka	173.8 ± 41.1	−2.7 (−6.2, 0.9)	0.1421	0.378	0.47 (0.42, 0.52)
INTERSALT1	152.0 ± 36.5	−24.5 (−28.0, −21.0)	<0.0001	0.410	0.48 (0.42,0.52)
INTERSALT2	149.7 ± 36.0	−26.8 (−30.1, −23.3)	<0.0001	0.409	0.47 (0.42,0.52)
Toft	182.4 ± 50.2	5.9 (2.3, 9.6)	0.0015	0.374	0.51 (0.46,0.55)
Whitton	152.3 ± 44.8	−24.2 (−27.7, −20.6)	<0.0001	0.393	0.50 (0.45,0.55)

24-h urinary sodium (24-hUNa), Person correlation coefficient (Person’s *r*), intraclass correlation coefficients (ICC), INTERSALT with potassium formula (INTERSALT1), INTERSALT without potassium formula (INTERSALT2), ^1^ Person correlation all *p* < 0.0001, ^2^ The value of single measures were used (all *p* < 0.01).

**Table 3 nutrients-12-00798-t003:** Misclassification of the six predicted formulas at individual level, *n* (%).

	Conversion of Salt Intake by 24-hUNa Excretion				Total (*n* = 1671)
	<9 g/day (*n* = 743)	9–11.99 g/day (*n* = 429)	12-14.99 g/day (*n* = 249)	≥15 g/day (*n* = 250)	
Kawasaki	613 (82.5)	302 (70.4)	184 (73.9)	101 (40.4)	1200 (71.8)
Tanaka	393 (52.9)	203 (47.3)	186 (74.7)	227 (90.8)	1009 (60.4)
INTERSALT1	209 (28.1)	269 (62.7)	227 (91.2)	245 (98.0)	941 (56.3)
INTERSALT2	196 (26.4)	281 (65.5)	230 (92.4)	245 (98.0)	952 (57.0)
Toft	365 (49.1)	287 (66.9)	186 (74.7)	180 (72.0)	1018 (60.9)
Whitton	228 (30.7)	263 (61.3)	219 (87.9)	238 (95.2)	948 (56.7)

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
