# Peer review of "Estimating 24-Hour Sodium Excretion from Spot Urine Samples in Chinese Adults: Can Spot Urine Substitute 24-Hour Urine Samples?"

_nutrients, 2020, doi:10.3390/nu12030798_

Round 1

Reviewer 1 Report

This manuscript validated the six predictive equations in estimating 24-h UNa excretion using spot urine samples in both population and individual level. The authors found that Tanaka method performed best in estimating mean population 24-h UNa, but all these six methods performed poorly at individual levels. It is reasonably well written and it is a good guide for researchers who want to participate in this field.

I have a minor comment for consideration, not suggestion.

24-h urinary sodium excretion critically depends on sodium intake, especially, at the day or the day before sodium intake. I can speculate that spot urine sampling just after 24-h urinary sodium collection could better represent the 24-h urinary sodium excretion than that of just before 24-h urinary sodium collection. I can also speculate that the sodium intake at the time of 24-h urinary collection was less than that of usual daily sodium intake, especially lower than that of just the day before collection, which might possibly affect the results. That may be one of the reasons why salt excretion in this population (10.2g/day) was somewhat lower than that of the general population in China (12-14g/day). But this is just my speculation.

(Trivial)

Figure legend of Figure 3. It should be Figure 3, not Figure 2.

Author Response

Point 1: 24-h urinary sodium excretion critically depends on sodium intake, especially, at the day or the day before sodium intake. I can speculate that spot urine sampling just after 24-h urinary sodium collection could better represent the 24-h urinary sodium excretion than that of just before 24-h urinary sodium collection. I can also speculate that the sodium intake at the time of 24-h urinary collection was less than that of usual daily sodium intake, especially lower than that of just the day before collection, which might possibly affect the results. That may be one of the reasons why salt excretion in this population (10.2g/day) was somewhat lower than that of the general population in China (12-14g/day). But this is just my speculation.

Response 1: Thank you very much for your comments. I agree with you that 24-h urinary sodium excretion for individual level is varied day to day critically depends on sodium intake before survey day. You mentioned that spot urine sampling just after 24-h urinary sodium collection could better, which is important question. We need to answer which is better using different time urine (first-morning urine, second morning urine, afternoon urine, evening urine or random time urine) to estimate 24-h urinary sodium excretion. Many studies are exploring this issue.

In our study, we collected the first-morning urine before 24-h urine collection, which is due to convenience for field investigation. 24-h urine collection began after participants emptied their bladders at investigation site. The last urine should be collected into the bottle when ending time. Therefore spot urine sampling just after 24-h urine collection is not convenience in large population survey.

You mentioned that the sodium intake at the time of 24-h urinary collection was less than that of usual daily sodium intake. I also think so. Further urine creatinine also varied daily because of dietary protein intake and physical activity. Considering these sensitive factors, we repeatedly emphasized that subjects should not change their daily life during our survey; however, it is difficult to ensure participant compliance. These are the quality control issues we need to consider in the field investigation.

Point 2: (Trivial) Figure legend of Figure 3. It should be Figure 3, not Figure 2.

Response 2: Done. We are very sorry for our negligence and I have revised the legend of figure 3.

Reviewer 2 Report

The authors describe a study in which they assess the effectiveness of 6 different calculations which can be applied to spot urine samples to estimate sodium excretion and, by extrapolation, intake. The authors specifically applied the calculations to the Chinese population to establish if they are appropriate for this group. Overall, the study shows very wide variation in estimates of sodium excretion across different methods, with only one of the calculations (Tanaka) showing any level of consistency and reliability. I felt on the whole that the article was well written and communicates a useful finding that is highly pertinent to the assessment of intake and status at the population level. I have one or two comments, alterations or requests for clarification which are listed below. 

Introduction. 

Line 50: Wording needs adjustment. 

Methods. 

Line 97: Need to include the questions that were asked to generate demographic data. This can be in the supplemental information, but it needs to be communicated. 

Results. 

Table 1: Are P values significances of independent t-tests between males and females? If so, this should be noted in the table legend.  

Table 2: Need to add units of measurement for 24 U Na excretion column. I presume this is data from all participants, so this should be communicated in the legend. I also think it would be helpful to add a row showing the 24 hr measurement so that the values can be easily compared. 

Figure 2: I didn’t really understand the full relevance of these figures and wasn’t sure how they added to the overall message of the paper. This may well be due to my own ignorance, but in the interests of others like myself, I would request a clearer clarification of what is presented and why. Details may be added in the figure legend. 

Author Response

Response to Reviewer 2 Comments

Point 1: Introduction. Line 50: Wording needs adjustment. 

Response 1: We are very sorry for our negligence and I have revised t that sentence.

Point 2: Methods. Line 97: Need to include the questions that were asked to generate demographic data. This can be in the supplemental information, but it needs to be communicated. 

Response 2: Thank you for your suggestion. However, I found that related socio-demographic and lifestyle factors were not used in the analysis in the articles. So I think add more information in the article may be not good. I am sorry for the misunderstanding due to unclear descriptions in my manuscript and I revised the descriptions.

I am very glad to share some detail information in questionnaire with you. A close-ended questionnaire was administered face-to-face by trained public health staff. We collected information on individual socio-demographic, self-reported history of hypertension and diabetes, as well as the lifestyle habits of smoking, alcohol use, physical activity, and diet. We also collected data on their knowledge of health outcomes of sodium intake and hypertension, perceptions of salt consumption, and attitudes and intentions toward reducing salt intake.

Individual Socio-demographic question main including:

Participant Name;

Address and contact details;

Gender;

Date of Birth;

ethnic groups;

Current Marriage Status;

Highest level of completed education;

Main occupation;

Current Medical Insurance Status.

Point 3: Results. Table 1: Are P values significances of independent t-tests between males and females? If so, this should be noted in the table legend.  

Response 3: Yes, you are right. I revised the table legend and statistical analysis description.

Point 4: Table 2: Need to add units of measurement for 24 U Na excretion column. I presume this is data from all participants, so this should be communicated in the legend. I also think it would be helpful to add a row showing the 24 hr measurement so that the values can be easily compared. 

 Response 4: Your suggestion is very important. I added units of measurement and a row showing the measured 24-h urinary sodium, revised the legend of table 2.

Point 5: Figure 2: I didn’t really understand the full relevance of these figures and wasn’t sure how they added to the overall message of the paper. This may well be due to my own ignorance, but in the interests of others like myself, I would request a clearer clarification of what is presented and why. Details may be added in the figure legend. 

Response 5: I am sorry for the misunderstanding due to unclear descriptions in Figure 2.

This is really hard to understand. For each participant, the relative differences was calculated by the following formula: [(predicted value-measured value)/measured value].

For example, one subject with measured value 10g/day, predicted salt intake using Kawasaki formula is 16 g/day, the relative difference is (16-10)/10=60%, which is classified  into beyond ±40% group. See figure 2, in Kawasaki bar, 51.0% participants (1671*51.0%=852 participants) relative difference is beyond ±40% intervals. It is showed predicted very inaccurate at individual level for Kawasaki formula.

Reviewer 3 Report

This manuscript focuses on the validation of using spot urine estimating 24-Hour Sodium Excretion in adults in Shandong Province, China.

The importance of estimating 24-h sodium excretion is well described in the manuscript. The authors are well aware of previous research works conducted in the field and the limitation of evaluating the day to day variation in salt intakes using a single 24-h UNa excretion. The authors also acknowledged that the urine samples were collected in Shandong Province only, which is not representative of the Chinese population.

There are several points to be considered:

Major points:

  • On page 2 line 71, regarding the exclusive criteria, samples with one or two missing collections were eligible for analysis, how to address this problem with potential under collection?

Could the authors provide the percentage of samples with missing collections?

  • Is there any adjustment on the measured sodium excretion upon different length of urine collection time, e.g., samples collected between 22 hours and 24 hours? Were the results considered as 24-h sodium excretion though the collection time was only 22 hours?
  • The way of determining misclassification of 4 groups could be improved. Considering a participant with a measured salt intake of 11.9 g/day and estimated results of 12.1 g/day, this participant will be misclassified to group of 12-14.99g/day when applying the classified rules in the manuscript.

An alternative could be, for a user with salt intake X g/day, predicted salt intake X’ being out of X ±1.5 g/day will be misclassified case.

Minor points:

  • Please provide further details about the sentence ‘This goal cannot be achieved in China by 2030’ on page 8.

Author Response

Response to Reviewer 3 Comments

Point 1: On page 2 line 71, regarding the exclusive criteria, samples with one or two missing collections were eligible for analysis, how to address this problem with potential under collection? Could the authors provide the percentage of samples with missing collections?

Response 1:  Regarding the exclusive criteria, we read a lot of literature and conducted some discussion. The completeness of 24-hour urine is key issue and very difficult. Samples with occasionally one missing collections were eligible in our study. I added this content into the limitations.

2043 participants completed 24-h urine sample. 284 (13.9%) participants were the ineligible urine, including the collection time was less than 22-h, or more than two urine samples were not collected, or participant recall regarding timing or missed collections was uncertain, or urine volume was <500 ml.

84 participants were excluded because urinary creatinine was not within ±2 standard deviation of the gender-specific mean.

4 participants missing SU samples were excluded.

So, the final analysis sample included 1671 participants.

Point 2: Is there any adjustment on the measured sodium excretion upon different length of urine collection time, e.g., samples collected between 22 hours and 24 hours? Were the results considered as 24-h sodium excretion though the collection time was only 22 hours?

Response 2: You question is very important. Frankly, the duration of urine collection between 22 to 24 hours were acceptable in the study, without any other adjustment. I think this is a limitations and added this content in my manuscript.

Point 3: The way of determining misclassification of 4 groups could be improved. Considering a participant with a measured salt intake of 11.9 g/day and estimated results of 12.1 g/day, this participant will be misclassified to group of 12-14.99g/day when applying the classified rules in the manuscript.

An alternative could be, for a user with salt intake X g/day, predicted salt intake X’ being out of X ±1.5 g/day will be misclassified case.

Response 3: You comment is very important. I think the information showed Figure 2 and Figure 3 can answer your question.

Absolute differences were analysed according to the formula: [absolute of (predicted value-measured value)]. For example, for a user with salt intake 10g/day, predicted salt intake 11g/day, Absolute differences is 1g/day, which will be classified into with ±17.1mmol/24-h (1 g/day salt).  In figure 3, Using Kawasaki formula, only 13.5% participants (about 226 participants) absolute differences with in ±1g/day. Similarly, the Figure 3 show that 52.3% participants’ absolute differences beyond ±68.4mmol/24-h (4 g/day salt), which indicted very inaccurate at individual level for Kawasaki formula.

According to your comment, I try to present more information by change the categories. Salt intake were classified into 4 categories: <7 g/day, 7-9.99 g/day, 10-12.99 g/day, ≥13 g/day. The results are shown in the table below.

I think this analysis provides more information, I put this table in the supplementary material.

Table S3. Misclassification of the six predicted formulas at individual level, n(%).

Conversion of salt intake by 24-hUNa excretion

Total

(n=1671)

<7 g/day

(n=423)

7-9.99 g/day (n=473)

10-12.99 g/day (n=373)

≥13 g/day (n=402)

Kawasaki

406 (96.0)

387 (81.8)

262 (70.2)

107(26.6)

1162 (69.5)

Tanaka

363 (85.5)

234 (49.5)

218 (58.4)

301 (74.9)

1116 (66.8)

INTERSALT1

284 (67.1)

204 (43.1)

265 (71.0)

359 (89.3)

1112 (66.5)

INTERSALT2

268 (63.4)

217 (45.9)

275 (73.7)

366 (91.0)

1126 (67.4)

Toft

400 (94.6)

205 (43.3)

275 (73.7)

241 (60.0)

1121 (67.1)

Whitton

256 (60.5)

254 (53.7)

268 (71.8)

349 (86.8)

1127 (67.4)

Point 4: Minor points: Please provide further details about the sentence ‘This goal cannot be achieved in China by 2030’ on page 8.

Response 4: I am sorry for writing inadequate sentences. I revised this s